Enriched environment prevents oxidative stress in zebrafish submitted to unpredictable chronic stress

Marcon Matheus 1
Mocelin Ricieri 1
Sachett Adrieli 1
Siebel Anna M. 2
Herrmann Ana P. 3
Piato Angelo angelopiato@ufrgs.br 1 3 4
1 Programa de Pós-graduação em Neurociências, Universidade Federal do Rio Grande do Sul , Porto Alegre , RS , Brazil
2 Programa de Pós-graduação em Ciências Ambientais, Universidade Comunitária da Região de Chapecó , Chapecó , SC , Brazil
3 Programa de Pós-graduação em Farmacologia e Terapêutica, Universidade Federal do Rio Grande do Sul , Porto Alegre , RS , Brazil
4 Zebrafish Neuroscience Research Consortium (ZNRC) , Los Angeles , United States of America
Maggi Laura
Electronic publication date: 2018 Jul 5
Publication date: 2018
Volume: 6
Electronic Location ID: e5136
Received 2018 Mar 10; Accepted 2018 Jun 8
Copyright: ©2018 Marcon et al.
Copyright year: 2018
Copyright holder: Marcon et al.
License: This is an open access article distributed under the terms of the Creative Commons Attribution License, which permits unrestricted use, distribution, reproduction and adaptation in any medium and for any purpose provided that it is properly attributed. For attribution, the original author(s), title, publication source (PeerJ) and either DOI or URL of the article must be cited.
License URL: https://creativecommons.org/licenses/by/4.0/

Keywords: Environmental enrichment, Unpredictable chronic stress, Oxidative stress, Zebrafish

Funding: Conselho Nacional de Desenvolvimento Científico e Tecnológico—Brazil 401162/2016-8 302800/2017-4 Coordenação de Aperfeiçoamento de Pessoal de Nível Superior (CAPES) This work was supported by Conselho Nacional de Desenvolvimento Científico e Tecnológico—Brazil (CNPq, Proc. 401162/2016-8 and 302800/2017-4). Adrieli Sachett, Ricieri Mocelin, and Matheus Marcon are recipients of fellowships from Coordenação de Aperfeiçoamento de Pessoal de Nível Superior (CAPES). The funders had no role in study design, data collection and analysis, decision to publish, or preparation of the manuscript.

==============================
Background

The enriched environment (EE) is a laboratory housing model that emerged from efforts to minimize the impact of environmental conditions on laboratory animals. Recently, we showed that EE promoted positive effects on behavior and cortisol levels in zebrafish submitted to the unpredictable chronic stress (UCS) protocol. Here, we expanded the characterization of the effects of UCS protocol by assessing parameters of oxidative status in the zebrafish brain and reveal that EE protects against the oxidative stress induced by chronic stress.

Methods

Zebrafish were exposed to EE (21 or 28 days) or standard housing conditions and subjected to the UCS protocol for seven days. Oxidative stress parameters (lipid peroxidation (TBARS), reactive oxygen species (ROS) levels, non-protein thiol (NPSH) and total thiol (SH) levels, superoxide dismutase (SOD) and catalase (CAT) activities were measured in brain homogenate.

Results

Our results revealed that UCS increased lipid peroxidation and ROS levels, while decreased NPSH levels and SOD activity, suggesting oxidative damage. EE for 28 days prevented all changes induced by the UCS protocol, and EE for 21 days prevented the alterations on NPSH levels, lipid peroxidation and ROS levels. Both EE for 21 or 28 days increased CAT activity.

Discussion

Our findings reinforce the idea that EE exerts neuromodulatory effects in the zebrafish brain. EE promoted positive effects as it helped maintain the redox homeostasis, which may reduce the susceptibility to stress and its oxidative impact.

Introduction

Currently, the issue of housing condition of animals in the laboratory is widely acknowledged in scientific discussions (Kempermann, Kuhn & Gage, 1997; National Research Council (US) Committee for the Update of the Guide for the Care and Use of Laboratory Animals, 2011; Kim et al., 2017). Sherwin (2004), for example, reviewed the effects of standard laboratory cages design and husbandry in rodents. The author argues that validity of research data is another reason to improve housing conditions of experimental animals, besides the more often mentioned welfare aspect. Different husbandry conditions could contribute to the observed data variability and irreproducibility among different laboratories. For zebrafish, however, studies comparing the impact of housing conditions on research outcomes are scarce. The standard laboratory condition for this species consists of housing the animal in shoals in barren tanks only with a recirculation system, heater thermostat (temperature control), and water in ideal conditions including physical, chemical and biological characteristics (pH, salinity, alkalinity, hardness, dissolved oxygen and nitrogen residues) (Lawrence & Mason, 2012). However, this housing environment is very far from the natural habitat conditions to zebrafish, that lives in shallow water with aquatic vegetation and gravel substrates (Arunachalam et al., 2013).

In this context, the practice of enriched environment (EE) arose from efforts to minimize the impact of environmental conditions on laboratory animals (Diamond, Krech & Rosenzweig, 1964; Kempermann, Kuhn & Gage, 1997; Young et al., 1999; Van Praag, Kempermann & Gage, 2000; Bennett et al., 2006). EE is a laboratory housing model that aims to approximate the housing condition to the natural habitat of the animals. This form of housing includes interventions that contributes to increase stimulation of sensory, motor and cognitive neuronal systems of the brain, and it allows or facilitates the animals to develop natural and species-specific behaviors (Van Praag, Kempermann & Gage, 2000; Lazarov et al., 2005; Meshi et al., 2006; Nithianantharajah & Hannan, 2006; Tanti et al., 2013).

A growing body of evidence reports the beneficial effects of EE for several species of laboratory animals (Nilsson et al., 1999; Brown et al., 2003; Sale et al., 2007; Sztainberg & Chen, 2010; Toth et al., 2011; Gapp et al., 2016). Studies in rodents report that EE has positive effects on the maintenance of the redox state, promoting protection against oxidative stress. For example, it was demonstrated that rats housed in EE presented reduced oxidative stress biomarkers, such as thiobarbituric acid reactive substances (TBARS), protein oxidation, superoxide anion (O2•−) activity and higher values for antioxidant parameters, such as the total radical antioxidant parameter, catalase (CAT) and superoxide dismutase (SOD) when compared to animals housed in standard laboratory conditions (Mármol et al., 2015) . A report showed that EE attenuated the upregulation of biomarkers of ROS production, such as levels of oxidase 2 (NOX2) and 8-hydroxy-2-deoxyguanosine (8-OH-dG) induced by a rat model of post-traumatic stress disorder (Sun et al., 2016). In addition, EE promoted neuroprotection through epigenetic mechanisms because it increased levels of DNA methylation and reduced levels of hydroxymethylation, as well as increased histone acetylation levels of H3 and H4. This resulted in increased expression of genes encoding oxidative machinery proteins, such as Hmox1, Aox1, and Cox2, and reduced expression of inflammatory genes such as IL-6 E Cxcl10 (Griñan Ferré et al., 2016).

Lastly, although some evidence suggests promising results for EE on oxidative stress in rodents, studies that report the effects of EE on this parameter in zebrafish are still scarce. Considering that in our previous study we demonstrated that EE prevented the increase of ROS in zebrafish submitted to the unpredictable chronic stress (UCS) protocol (Marcon et al., 2018), we hypothesized that zebrafish submitted to UCS protocol housed in EE would be less vulnerable to oxidative stress. Therefore, in this study we tested the effects of EE in zebrafish submitted to UCS on a range of oxidative stress parameters including lipid peroxidation (TBARS), reactive oxygen species (ROS), non-protein thiol (NPSH) levels, total thiol (SH) levels, superoxide dismutase (SOD) activity, and catalase (CAT) activity.

Material and Methods

Animals

A total of 150 short fin wild-type (WT) adult zebrafish (Danio rerio) 50:50 male/female ratio over 6-month-old were purchased from Delphis aquariums (Porto Alegre, Brazil). The fish were kept in a closed acclimation tank system of 16 L (40 × 20 × 24 cm, <2 fish per liter) for two weeks. Tanks were filled with non-chlorinated tap water, well-aerated in appropriate conditions as previously reported by Marcon et al. (2018). The illumination of the room was 14/10 h light/dark photoperiod cycle (lights on at 06:00 am). The fish were fed twice a day with a commercial flake fish food (Alcon BASIC®, Alcon, Brazil) and nauplii of brine shrimp (Artemia salina). The amount of food was calculated based on the number of fish per tank and followed the instructions of the Zebrafish Book (Westerfield, 2000). All experiments were approved by the Ethics Committee of Universidade Federal do Rio Grande do Sul (#30992/2015).

Experimental procedures

EE methodology followed that described by Marcon et al. (2018) and is shown in Figs. 1 and 2. After the acclimation period (two weeks), zebrafish were randomly assigned to one of two experimental housing environments: barren tank (BARREN) or enriched environment (EE). BARREN condition consists of standard laboratory tank as described above and containing only water, heater, filter, and aeration system while EE condition consists the same BARREN condition plus tank gravel in the bottom (English sea stones, 4–9 mm, 3 cm high from the bottom of the tank), a ruin-like plastic object, and three submerged plastic plants (two 10 cm tall and one 20 cm tall) (Fig. 1). All tanks of both experimental acclimation conditions were kept in a horizontal plane at the same room, so we used a white frosted cardboard (30 × 60 cm) placed only in between tanks to prevent visual contact of fish from different tanks.

Figure 1 Housing conditions.

(A) Barren tank; (B) Enriched environment tank. Image credit/source: (A and B) Matheus Marcon.

Figure 2 Experimental design.

The fish were housed in barren tank (BARREN) or enriched environment tank (EE) for 21 or 28 days. In the last seven days of the experimental protocol, they were submitted to the unpredictable chronic stress (UCS) protocol or remained unchanged (Control). The day after the last stressor, at 08:00 a.m., the fish were euthanized for biochemical analyzes of oxidative stress (the brain was used for the dosage of the brain was used for the dosage of lipid peroxidation (TBARS), reactive oxygen species (ROS) levels, non-protein thiol (NPSH) and total thiol (SH) levels, superoxide dismutase (SOD) and catalase (CAT) activities.

Zebrafish were kept in barren tank (BARREN) or enriched environment tank for 21 (EE 21) or 28 (EE 28) days. In both housing conditions, in the last seven days of the experimental protocol the animals were again divided into two experimental subgroups (non-stressed or stressed, respectively, S− and S+). The S+ groups were submitted to unpredictable chronic stress (UCS) protocol detailed bellow. At the end of the experimental protocol on day 29th (24 h of the last intervention), the animals were removed from their tanks by using a net and immediately anesthetized by rapid cooling (immersion in water at 2–4 °C). After cessation of opercular movements zebrafish were euthanized by decapitation. The brain was used for analysis of oxidative stress.

Unpredictable chronic stress (UCS) protocol

To induce stress in the zebrafish a UCS protocol was used which is already well established and described in our previous studies (Piato et al., 2011; Marcon et al., 2016; Rambo et al., 2017). UCS protocol was based on the following stressors (1) heating tank water up to 33 °C (30 min); (2) cooling tank water to 23 °C (30 min); (3) crowding of 12 animals in a 300-mL beaker (50 min); (4) transferring the animals to other tank with low water level exposing the dorsal body wall (2 min); (5) tank change, three consecutive times with 30-min interval; and (6) chasing with a net (8 min) that were randomly presented twice a day for 7 consecutive days (day 21 to day 28) to the animals of the stressed groups. The non-stressed (S−) animals were maintained in the same room and did not undisturbed throughout the experiments.

Oxidative stress analysis

For brain tissue preparation, immediately after euthanasia, fish were dissected out in ice and to each sample used five brains were pooled that were gently homogenized in ice-cold phosphate buffered saline (PBS; Sigma Aldrich, St. Louis, MO, USA) pH 7.4 and centrifuged at 10,000× g for 5 min at 4 °C to remove cellular debris. The supernatants were collected and used for estimation biomarkers associated with oxidation mechanisms and induction of oxidative stress described herein.

Lipid peroxidation (TBARS)

Lipid peroxidation (Draper & Hadley, 1990) was estimated by monitoring thiobarbituric acid reactive substance (TBARS) production. Briefly, a volume equal to 50–70 µg protein of brain homogenate were added to 150 µL of 2% trichloroacetic acid (TCA, Sigma Aldrich®) and centrifuged (10,000× g, 10 min). The supernatants were collected, mixed with 150 µL of 0.5% thiobarbituric acid (TBA; Sigma Aldrich, St. Louis, MO, USA) and then heated at 100 °C for 30 min. The reading of TBARS levels occurred in the microplate reader in absorbance at 532 nm, using 1,1,3,3-tetramethoxypropane (TMP; Sigma Aldrich, St. Louis, MO, USA) as a standard. Results were expressed as nanomoles (nmol) MDA/mg protein (n = 5).

Reactive oxygen species (ROS) assay

To evaluate the free radical content (LeBel et al., 1990; Ali, LeBel & Bondy, 1992), the fluorescent probe 2′, 7′-dichlorofluorescin diacetate (DCFH-DA; Sigma Aldrich, St. Louis, MO, USA) was used. Briefly, 25 µL of brain homogenate was incubated with of 1 mM DCFH-DA and PBS buffer at 37 °C for 30 min. ROS levels was estimated in the microplate reader in fluorescence at 520 nm of emission and 480 nm of excitation using dichlorofluorescein (DCF) as standard. Results were expressed as relative fluorescence unit (RFU) (n = 5).

Non-protein thiol (NPSH) levels

To estimate NPSH levels (Ellman, 1959), equal volumes (30 µL) of brain preparation and 6% trichloroacetic acid (Sigma Aldrich, St. Louis, MO, USA) was mixed and centrifuged (3,000× g, 10 min at 4°C). Subsequently, an aliquot of supernatant (50 µg protein) was further mixed with 10 mM 5,5-dithio-bis-2-nitrobenzoic acid (DTNB; Sigma Aldrich, St. Louis, MO, USA) dissolved in ethanol and the intense yellow color developed was measured in the microplate reader at 412 nm after 1 h of incubation at room temperature. Results were expressed as µmol NPSH/mg of protein (n = 5).

Total thiol (SH) levels

To estimate SH levels (Ellman, 1959), a volume equal to 50 µg protein of brain homogenate was mixed with 10 mM 5,5-dithio-bis-2-nitrobenzoic acid (DTNB; Sigma Aldrich, St. Louis, MO, USA) dissolved in ethanol (Sigma Aldrich, St. Louis, MO, USA). The intense yellow color developed was measured in the microplate reader at 412 nm after 1 h of incubation at room temperature. Results were expressed as µmol SH/mg of protein (n = 5).

Superoxide dismutase (SOD) activity

SOD activity (Misra & Fridovich, 1972) was estimated by quantifying the inhibition of superoxide-dependent adrenaline auto-oxidation. Adrenochrome formation rate was observed at 480 nm in the microplate reader in a reaction medium containing glycine-NaOH (50 mM, pH 10, Sigma Aldrich®), epinephrine (60 mM, pH 1.7; Sigma Aldrich, St. Louis, MO, USA), and homogenate brain (15–30–60 µg of protein). Results were expressed in Units/mg protein (n = 5).

Catalase (CAT) activity

CAT activity (Aebi, 1984) was estimated by measuring the rate of decrease in hydrogen peroxide (H2O2) absorbance at 240 nm. Assay mixture consisted of potassium phosphate buffer (Sigma Aldrich®), H2O2 (1 M; Sigma Aldrich, St. Louis, MO, USA) and brain homogenate (30 µg protein). The results were expressed in Units/mg protein (n = 5).

Protein determination

Protein was determined by the Coomassie blue method (Bradford, 1976) using bovine serum albumin (Sigma Aldrich, St. Louis, MO, USA) as standard. Absorbance of samples was measured at 595 nm.

Statistical analysis

Kolmogorov–Smirnov and Levene tests were used to determine the normal distribution of the data and homogeneity of variance, respectively. Results were analyzed by two-way ANOVA (stress and enriched environment as independent factors) followed by Tukey post hoc test for comparisons within groups and between housing conditions. Differences were considered significant at p < 0.05. The data were expressed as a mean + standard error of the mean (S.E.M.).

Results

Figure 3 shows the effects of EE on biochemical parameters associated with oxidative stress (TBARS and ROS) in zebrafish submitted to UCS and summarizes the two-way ANOVA analyzes. Regarding TBARS (Fig. 3A), two-way ANOVA revealed that UCS interacted with EE: stress only increased TBARS levels when fish were housed in barren tanks, but not when they were housed for 21 or 28 days of EE. Regarding ROS (Fig. 3B), two-way ANOVA also revealed an interaction between UCS and EE: increased ROS levels were observed only in stressed fish from barren tanks, but not from enriched tanks.

Figure 3 Effects of enriched environment for 21 (EE 21) or 28 days (EE 28) on biochemical parameters associated with oxidative stress in zebrafish brain submitted to unpredictable chronic stress (S+) or not (S−).

Lipid peroxidation (TBARS) and reactive oxygen species (ROS) levels. BARREN: barren tank. Data are expressed as a mean + S.E.M. n = 5. Two-way ANOVA/Tukey.

Figures 4 and 5 show the effects of EE on biochemical parameters associated with antioxidant mechanisms (NPSH and SH levels, SOD and CAT activity) in zebrafish submitted to UCS. In Fig. 4A, two-way ANOVA revealed an interaction between UCS and EE for NPSH levels: stress decreased NPSH levels only in fish from barren tanks, while EE for 21 or 28 days prevented this effect of stress. Regarding SH levels (Fig. 4B), two-way ANOVA revealed an interaction but no main effects of UCS and EE; post hoc tests, however, did not reach significance for multiple comparisons between groups.

Figure 4 Effects of enriched environment for 21 (EE 21) or 28 days (EE 28) on antioxidant mechanisms in zebrafish brain submitted to unpredictable chronic stress (S+) or no (S−). Non-protein thiols (NPSH) and Total thiol (SH) levels.

BARREN: barren tank. Data are expressed as a mean + S.E.M. n = 5. Two-way ANOVA/Tukey.

Figure 5 Effects of enriched environment for 21 (EE 21) or 28 days (EE 28) on antioxidant mechanisms in zebrafish brain submitted to unpredictable chronic stress (S+) or no (S−).

Superoxide dismutase (SOD) and Catalase (CAT) activity. BARREN: barren tank. Data are expressed as a mean + S.E.M. n = 5. Two-way ANOVA/Tukey.

In Fig. 5A, two-way ANOVA for SOD activity revealed an interaction between UCS and EE: stress decreased SOD activity, which was prevented only when fish were housed in EE for 28, but not 21 days. Regarding CAT activity (Fig. 5B), two-way ANOVA revealed main effects for UCS and EE, but no interaction between these factors; overall, stress decreased while EE increased CAT activity. Previously, some studies had already reported the protective potential of EE on maintenance of redox homeostasis. EE showed to prevent DNA oxidation (Kang et al., 2016; Sun et al., 2016), the increase of carbonyl protein (Herring et al., 2008), the increase of total free radicals content (Cechetti et al., 2012) and the increase of lipid peroxidation (Muhammad et al., 2017) in rodents.

Discussion

In this study, we replicate a previous result and expand the characterization of the effects of EE on mechanisms associated with antioxidant defenses and oxidative stress in zebrafish submitted to UCS. We demonstrated for the first time that UCS protocol induced several changes in redox homeostasis in the zebrafish brain and revealed that EE has a protective effect against the oxidative stress induced by the UCS protocol.

UCS protocol induces several biochemical changes in the zebrafish brain and through sustained activation of the neuroendocrine axis leads to increased cortisol levels (Piato et al., 2011; Manuel et al., 2014; Marcon et al., 2016; Rambo et al., 2017; Song et al., 2017). This was confirmed by the results recently published in our previous study, which showed that the UCS protocol increased cortisol levels while EE for 21 or 28 days prevented this increase (Marcon et al., 2018). In this way, the response to sustained stress leads to great energy expenditure and for this reason some cellular metabolic processes are accelerated (Otte et al., 2016), such as oxidative phosphorylation (Zorov, Juhaszova & Sollott, 2014) and β-oxidation of fatty acids (Carracedo, Cantley & Pandolfi, 2013). As a consequence, the excessive production of ROS can reach levels above the antioxidant defense capacity of the organism and consequently oxidize cellular structures leading to oxidative stress (Sies, Berndt & Jones, 2017; Poprac et al., 2017).

Particular features of nervous tissue, such as neurotransmitters metabolism, high iron content, low antioxidant capacity, neuronal membrane rich in polyunsaturated fatty acids and high oxygen consumption, make the brain an organ extremely susceptible to oxidative stress (Clarke & Sokoloff, 1999; Halliwell, 2006). Therefore, sustained antioxidant mechanisms are necessary for the maintenance of cerebral homeostasis (Finkel & Holbrook, 2000).

Glutathione (GSH) is the main antioxidant component in brain tissue. It is essential for maintenance of redox homeostasis, serving as the cofactor of the enzymes glutathione peroxidase (GPx) and glutathione-S-transferase (GST) and a direct neutralizer of ROS (Dringen, 2000; Dringen & Hirrlinger, 2003). Here we showed that the UCS protocol decreased NPSH levels, a measure that reflects the levels of GSH, while it did not alter the SH levels (thiols groups associated with cysteine residues), suggesting that chronic stress promoted the depletion of cerebral GSH in zebrafish.

At the same time, the UCS protocol decreased SOD but did not alter CAT activities. Physiologically the SOD enzyme plays a key role in neutralizing the superoxide anion (O2•−) to hydrogen peroxide (H2O2), which is synergistically converted to water (H2O) and oxygen (O2) by CAT (Fukai & Ushio-Fukai, 2011). Therefore, we hypothesized the decrease in the SOD activity induced by UCS may led to an excessive accumulation of O2•−. O2•− in high concentrations may contribute to oxidative stress through direct or indirect damage, for example, by the formation of other reactive species, such as peroxynitrite (ONOO−), H2O2 or hydroxyl radical (OH•) (Fenton reaction) (Pacher, Beckman & Liaudet, 2007). Additionally, we revealed here that the UCS protocol increases ROS production and therefore we suggest that the high levels of ROS associated with the decrease of the antioxidant mechanisms (GSH level and decreased SOD activity) led to an imbalance between its production and detoxification and consequently increased lipid peroxidation in stressed animals leading to oxidative stress. This is according to a previous study that showed a decrease in the values of total antioxidant status, SOD activity and the increase of lipid peroxidation in mice submitted to chronic unpredictable mild stress (Biala et al., 2017).

Oxidative stress is related to the development of mental disorders (Ng et al., 2008) and it was demonstrated to contribute to the pathophysiology of neurodegenerative diseases (Christen, 2000). Therefore, it is remarkable the need for studies that bring new discoveries in this line. Interestingly, here, we have shown for the first time that EE promoted protection against oxidative stress induced by UCS. We report that EE for 28 days prevented all changes induced by UCS in the oxidative status while EE for 21 days prevented the decreased of NPSH levels and the increased of the lipid peroxidation and ROS levels. Besides, both EE for 21 or 28 days increased the CAT activity.

In this study, we suggest that EE prevented the oxidative by preventing the decrease of antioxidant defenses (GSH level and SOD enzyme activity), as well as the increase of ROS levels. Furthermore, EE increases the expression of glucocorticoid receptors (Shilpa et al., 2017), which is associated with downregulation of neuroendocrine axis activity; this occurs by negative feedback at the cortisol receptor and reduces the response to sustained stress.

Conclusion

Our findings are in agreement with our previous study and together with the literature findings reinforce the idea that EE exerts neuromodulatory effects. Here, we revealed that EE promoted positive effects in the maintenance of redox homeostasis, which may reduce the susceptibility to stress and its oxidative impact. However, our data are still preliminary and require further investigation to establish and clarify the exact neurobiological mechanisms by which EE prevents changes in oxidative status. Also, we reinforce and suggest that zebrafish is a suitable animal model to investigate the neurobiology of stress and the effects of EE.

Supplemental Information

Data S1 Raw data

Click here for additional data file.

Additional Information and Declarations

Competing Interests

Author Contributions

Animal Ethics

Data Availability

Angelo Piato is an Academic Editor for PeerJ.

Matheus Marcon conceived and designed the experiments, performed the experiments, analyzed the data, prepared figures and/or tables, authored or reviewed drafts of the paper, approved the final draft.

Ricieri Mocelin and Adrieli Sachett performed the experiments, approved the final draft.

Anna M. Siebel contributed reagents/materials/analysis tools, approved the final draft.

Ana P. Herrmann conceived and designed the experiments, analyzed the data, prepared figures and/or tables, authored or reviewed drafts of the paper, approved the final draft.

Angelo Piato conceived and designed the experiments, analyzed the data, contributed reagents/materials/analysis tools, prepared figures and/or tables, authored or reviewed drafts of the paper, approved the final draft.

The following information was supplied relating to ethical approvals (i.e., approving body and any reference numbers):

All experiments were approved by the Ethics Committee of Universidade Federal do Rio Grande do Sul (#30992/2015).

The following information was supplied regarding data availability:

The raw data are provided in Data S1.

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
