# Peer review of "Enriched environment prevents oxidative stress in zebrafish submitted to unpredictable chronic stress"

_PeerJ, doi:10.7717/peerj.5136_

## Round 0.1 · original submission · Minor Revisions

One of the main concern of the reviewers is the cortisol quantification. Please, repeat this experiments or, at least, mention the previously published result.

I would suggest improving the result section.

Please also respond appropriately to all other reviewer comments.

Reviewer 1 ·

Basic reporting

No comment

Experimental design

no comment

Validity of the findings

To validate the hypothesis I think that the author should quantify the cortisol levels in the zebrafish

Additional comments

Perhaps, the author theory about the EE could increase GR as consecuence of downregulation of neuroendocrine axis activity could be evidence by determination of cortisol levels and GR .

Reviewer 2 ·

Basic reporting

The authors here report the contribution of the environmental conditions for Zebrafish exposed to unpredictable chronic stress (UCS) protocols. They characterize this impact using a set of different oxidative stress biomarkers. The paper is well written and easy to follow.

Experimental design

The experimental procedure is well explain both along the text and within the figure. No major concerns relate to animal care and the access to the raw data allows a straightforward independent statistical reanalysis if desired.

Validity of the findings

The results are interesting and highlight a potential byas in the results obtained by other researchers working with Zebrafish (or other models), who did not pay attention to the housing conditions. The references from the introduction show the impact of housing on oxidative stress in other models, but my impression is that the paper would really highlight its importance if the authors are capable cite an example of how this fact led to a missinterpretation of the results. Are there any reports on this line?

·

Basic reporting

The manuscript is well presented and the language on the whole is good. A couple of areas that would benefit from some clarification in terms of sentence construction are:

line 71. water quality. This needs and additional word to qualify what is meant - optimal water quality, defined water quality, water quality monitoring?

lines 99-102. This appears to be a circular argument and needs a bit of redrafting.

lines 157-160. This needs a little restructuring

line 179. change was to were

The background to the study was well explained with appropriate references and the structure and figures were well presented.

Experimental design

The experimental design is robust and well explained as was the research question. Ethical approval was obtained and the technical approach was sound.

Validity of the findings

I really liked this study and think it is both timely and important.Overall the data appear to be robust and well explained. I do have some queries around the interpretation of the data as outlined below
.
TBARS - all seems fine
ROS - ROS is observed to increase in the stressed, barren fish. EE21 (non-stressed), ROS is the same as barren but when stressed, ROS falls considerably compared to non-stressed. Can this be explained?
SH - SH levels falls following stress in the barren fish but increases in the EE fish although apparently coming from a lower starting point. It isn't clear if there is any statistical difference in this data although in the text it is stated that EE prevents an effect (although there isn't an observed effect in the barren group).
SOD - OK but is the effect at 28 days significant?
CAT - there is no effect on CAT in the barren group but EE increase CAT. However, stress seems to have an effect on CAT at 28 days - is this significant?

The conclusions presented in lines 297-299 I don't think can be drawn from the data presented.

Additional comments

I really liked this manuscript and think it is a great piece of work. I have outlined a few queries regarding the interpretation of the data. This might arise from the statistics not being described in the text and perhaps a table showing all the significant / no-significant comparisons would be useful.

Reviewer 4 ·

Basic reporting

The reporting of methods and data as well as intro/discussion was clear.

My concern is with the results section. While the data is clearly presented in the figures it is not clearly described in the Results. This section should receive scientific and language editing.

Experimental design

A clear extension of a previous study using the same model. It is original research that addresses biochemical changes in the brain in response to an environmental manipulation. The behavioural and biochemical methods were well described.

Validity of the findings

Consistent with their previous work, EE mitigate some but not all of the deleterious effects of the chronic stress-paradigmn.

Additional comments

a few comments on the text in the introduction are attached.

Annotated reviews are not available for download in order to protect the identity of reviewers who chose to remain anonymous.

---

## Round 0.2 · accepted · Accept

I am pleased to report that the paper has been considered suitable for publication, although not all reviewers have revised the final version.

# Reviewer 4 ·

Basic reporting

revised manuscript is acceptable for publication

Experimental design

revised manuscript is acceptable for publication

Validity of the findings

revised manuscript is acceptable for publication

Additional comments

revised manuscript is acceptable for publication